# Implementation Determinants of Knowledge Mobilization within a Quebec Municipality to Improve Universal Accessibility

**DOI:** 10.3390/ijerph192214651

**Published:** 2022-11-08

**Authors:** Maëlle Corcuff, François Routhier, Stéphanie Gamache, David Fiset, Jean Leblond, Marie-Eve Lamontagne

**Affiliations:** 1Center for Interdisciplinary Research in Rehabilitation and Social Integration, Centre Intégré Universitaire en Santé et Services Sociaux de la Capitale-Nationale, 525 Boul. Wilfrid-Hamel, Québec, QC G1M 2S8, Canada; 2Rehabilitation Department, Faculty of Medicine, Université Laval, Québec, QC G1V 0A6, Canada

**Keywords:** universal accessibility, human rights, municipality, local government, Convention on Rights of People with Disabilities, knowledge mobilization

## Abstract

According to the UN-CRPD, cities must develop action plans about universal accessibility (UA). Operationalization of these plans is complex, and little is known about what municipal employees know about UA. Aim: The aim is to document implementation determinants of UA within a municipal organization in Quebec, Canada. Methods: An observational cross-sectional study was performed. Employees answered a survey based on the TDF and the DIBQ. Facilitators, barriers, and factors influencing the determinants were identified. Results: A total of 43% of the employees completed the survey. The implementation of UA measures is more facilitated by their beliefs about the impact on citizens, while the external context hinders the proper implementation. It is also influenced by six factors: (1) professional role, (2) capacity, (3) resources, (4) willingness, (5) characteristics, and (6) feedback. Discussion: Results suggest that understanding the consequences, sufficient resources, abilities, and willingness can influence implementation of UA. Conclusion: These findings have informed the objectives of the next action plan of the municipal organization and could guide the development of solutions.

## 1. Introduction

In the Convention on the Rights of Persons with Disabilities (CRPD) adopted by the United Nations (UN) in 2006, universal accessibility is recognized as a fundamental human right and is essential to good health and quality of life of every individual [1,2]. Universal accessibility is defined as “the character of an environmental design that allows all individuals to carry out their activities independently, with equity and in an inclusive approach” [3]. Universal accessibility aims to create inclusive and adapted physical environments for the entire population such that inclusive and equitable environments would provide people with autonomy to practice their occupations [4]. It also seeks to make our built environment as enabling as possible to facilitate the optimal achievement of our objectives [5]. Through universal accessibility, all individuals (e.g., older adults, families, people with disabilities, and tourists) can have access to buildings, services, resources, and activities [6]. Indeed, more and more people are starting to find accessibility positively impacting their lives, whether it is the convenience of automatic doors when you have your hands full, elevators for sick or injured people, or even ramps and curbs, which are greatly appreciated by skateboarders [5]. However, the scientific literature shows that people living with disabilities experience more daily obstacles related to their environment and would benefit from actions taken to improve universal accessibility [4,7,8,9,10]. The attention paid to the concept of universal accessibility has recently grown and is materialized through the improvement and the implementation of new solutions and services in municipal organizations. People would benefit from fair access to environments, services, and municipal installations that are safe, healthy, and adapted to all [11,12,13].

Article 9 of the CRPD stipulates that “States Parties shall take appropriate measures to ensure to persons with disabilities access, on an equal basis with others, to the physical environment, to transportation, to information and communications, including information and communications technologies and systems, and to other facilities and services open or provided to the public, both in urban and in rural areas” [1]. More than 120 countries have developed frameworks defining universal accessibility practices and measures (https://www.un.org/development/desa/disabilities/disability-laws-and-acts-by-country-area.html, accessed on 20 January 2021). For example, Canada committed to establish accessibility measures and standards through the Accessible Canada Act [14]. This Act aims to ensure the economic, social, and civic participation of all people in Canada, regardless of their disabilities, and to enable them to fully exercise their rights and responsibilities as citizens in a country without obstacles. From a local perspective, in 2004 the province of Quebec (Canada) adopted the Act to secure handicapped persons in the exercise of their rights with a view to achieving social, school and workplace integration [15]. Article 61.1 of this Act stipulates that each city with more than 15,000 citizens must have an action plan identifying obstacles to the integration of people with disabilities, and describe the measures taken and planned in order to reduce them in sectors of activity such as employment, culture, or transportation [15]. Seventy-three cities in the province have created such an action plan and policies to ensure the implementation and improve actions towards universal accessibility [16].

Although there is legal access, there is still barriers in the environment, especially for people with disabilities [5]. While municipal administrations in Canada have developed these action plans, we do not know what is understood or what knowledge city employees gained about the concept of universal accessibility. Given the large number of cities in the province of Quebec, this research will promote the implementation of universal accessibility in cities. The lack of literature about employees’ knowledge, perceptions, feelings, or reactions in relation to the concept of universal accessibility and their influence on their daily working activities limit the capacity to purposefully improve actions to support cities in the implementation of universal accessibility measures and services. The primary aim of this research is to document the determinants (facilitators and barriers) influencing the implementation of universal accessibility measures among the municipal organization’s employees. The secondary aim is to identify the important aspects required to identify actions or factors that should be considered in the next action plan regarding universal accessibility.

## 2. Materials and Methods

### 2.1. Study Design

We performed an observational cross-sectional study. We were guided by the Theoretical Domain Framework model (TDF) throughout the research process [17]. This framework used in health and psychological research [18,19,20] allows the identification of a set of behaviour–change domains for implementation research to better understand the change process for the use of a given innovation [17,21]. The TDF documents 10 determinants (knowledge, skills, social/professional role, beliefs about capabilities, beliefs about consequences, intentions, innovation, internal and external context, and social influences).

### 2.2. Participants

A large metropolitan city of approximately 550,000 citizens located in the province of Quebec, Canada, expressed a need to document and evaluate strategies used in knowledge mobilization regarding universal accessibility to optimize and improve their next action plan. This city was chosen for this research because several employees and administrative units are involved in the implementation of universal accessibility and the research team was contacted to assist to provide relevant and useful evidence-based strategies. A stratified sampling was used, based on a contact list of the municipal organization’s administration’s employees from 17 administrative units concerned by universal accessibility, such as various districts, culture, heritage and international relations, development and environmental planning, fire protection, transportation and smart mobility, emergency services office, major events office, or building management. All administrative units were solicited to have a global portrait of the municipal employees, whether they were in direct contact with citizens or not. Over 600 employees were contacted by email. The inclusion criteria were to: (1) have worked for the organization for a minimum of one year and (2) work at least 14 h per week. These inclusion criteria were decided to be sure to collect the opinion of employees who are well integrated in the organization. Stratified sampling aimed to randomly select employees while respecting these two conditions: (1) all job categories had to be represented in each of the administrative units (seasonal, manual, professionals, public agent, and managerial) and (2) if the administrative unit had less than 50 employees, all employees of this administrative unit were surveyed. These two conditions were to maintain representation of all municipal employees.

### 2.3. Procedures

The first contact was completed via an email sent by the office of the assistant executive director. All 17 managers units were contacted and asked to encourage their employees to complete the survey. Managers were then asked to share the information with their team members and encourage them to complete the survey. The email indicated the nature and goals of the study as well as the participants’ expected contribution. A link provided in that email led to the survey’s homepage, where the entire study was presented to then further obtain free and informed consent. By clicking on “Take Survey” at the end of the page, they consented to the study. Participants had two weeks to complete the survey, on their working hours. One follow-up was made one month later.

### 2.4. Measures

Determinants influencing the implementation of universal accessibility were explored using a quantitative online survey based on the TDF. The survey was created based on the Determinants of Implementation Behaviour Questionnaire (DIBQ) [22], which is related to the TDF framework from Michie (2005). To adapt the DIBQ to the municipal context, questions were selected by the project’s committee, composed of five representatives of the municipal organization administration, two researchers (MEL and FR), a research professional (DF), and a PhD student (MC). The nature of the universal accessibility concept and the context of a municipal public organization were specified. It was then validated by the project’s committee, and the survey was created in an electronic survey platform, commonly used by the city. For the questions regarding knowledge of politics, the participant had to rate his/her level of knowledge on a 4-point Likert scale (0 = I do not know it; 1 = I know it exists; 2 = I know its content; and 3 = I use it in my work). For all ten determinants, they had to indicate how much they felt concerned by affirmations on a 4-point Likert scale (0 = not at all, 1 = a little, 2 = moderately, and 3 = a lot). The use of a 4-level Likert scale was intended to prevent participants from being tempted to choose the intermediate (neutral) option [23,24]. Annex 1 exposes the definition of each determinant and the survey questions associated.

### 2.5. Analysis

The quantitative data were extracted from the electronic platform. The data were summarized descriptively according to the determinants to explore which were the facilitators and barriers to knowledge mobilization. If more than half of the employees answered in favour of the question (2 = moderately or 3 = a lot), this was interpreted as a facilitator, whereas if the majority answered against the question (0 = not at all or 1 = a little), this was interpreted as a barrier. A categorical principal components analysis (CATPCA) was conducted using SPSS 25 to identify factors influencing the observed determinants. Given the large number of variables, this type of factorial analysis reduced the amount of data necessary to identify factors that may explain the majority of the observed variance [25,26]. The CATPCA is a flexible alternative to group a set of variables under one or more factors, that is adapted for ordinal variables that are not linearly related to each other [25,27]. The 19-item questionnaire, adapted from the DIBQ, was built according to the TDF framework of Michie (2005) that targets ten concepts or components. To empirically verify how these components were present, independently or not, within the sample, a principal components analysis was performed. Since answers were drawn from Likert scales, the SPSS routine was adapted for ordinal data (CATPCA). These factors were discussed and identified between the authors (MC, FR, SG, JL, and MEL). We could benefit from the expertise in factorial analysis (JL) and in implementation frameworks (MEL).

Two items of the questionnaire were not included in this analysis because they did not show variance. These items are from the beliefs on consequences determinants: “For me, it is important that the city provide services and facilities that meet the needs of people with disabilities and older adults” and “If I provide municipal services and environments that meet the needs of people with disabilities and seniors, citizens will appreciate it”. As a result, 17 items were included in the CAPTCA analysis to identify factors influencing the implementation of universal accessibility measures from one person to the other.

## 3. Results

A total of 268 municipal employees (43% of the ~600 employees) completed the survey. The sample was composed of 137 men and 127 women (4 indicated no gender), 72.0% of whom were aged between 41 and 60 years. The employees held different types of job: seasonal (*n* = 3), manual (*n* = 20), professionals (*n* = 86), public agent (*n* = 109), and managerial (*n* = 50). 

Table 1 shows the descriptive statistics that emerged from the survey regarding the different TDF domains of implementation among municipal employees. Overall, 84.2% of the participants believed that the municipal services and infrastructure are moderately or very accessible for people with disabilities and older adults. In addition, 60.4% expressed that they moderately or very much take into account the needs of people with disabilities and the elderly in their daily operations. However, the results also show little knowledge of the provincial law, the government policy, the municipal action plan, and the city’s practical guide to universal accessibility. Indeed, 11.5% of participants said they know the content or use the action plan in their work, while half of them (50.9%) reported not knowing its existence.

### 3.1. Facilitators to the Implementation of Universal Accessibility Measures

A total of 90.6% of participants said that it is very important that the city offers services and equipment that meet the needs of people with disabilities and the elderly and 84.0% believed very much that if they offer municipal services and environments that meet the needs of people with disabilities and the elderly, citizens will appreciate it. Additionally, 68.4% believed in their capacities to implement the measures in the action plan and think it is moderately or very easy for them to carry out the actions of their unit included in the municipal action plan. The answers showed that 51.0% of participants considered that it is very much their responsibility, as a municipal employee, to provide municipal services and environments that meet the needs of people with disabilities and the elderly and 61.9% have very much the intention to offer accessible municipal services in line with the municipal action plan for universal accessibility in the coming year. Manager support, classified in the determinant referring to the intern context, is also expressed as a facilitator. Among the participants who answered this question (*n* = 141), 51.8% said they have a lot of support from their manager and 28.4% said they have moderate support.

### 3.2. Obstacles to the Implementation of Universal Accessibility Measures

The results of the survey showed that the external context limits the capacity of employees to apply universal accessibility measures. The external context includes material and financial resources as well as available tools. In fact, 54.0% of participants said they do not have or have little material resources and 61.8% said they do not have or have little financial support to provide municipal services and environments that meet the needs of people with disabilities and the elderly. Additionally, more than half of the employees (56.9%) reported that their employer does not provide enough material, tools, or documentation to offer adequate services. Except for the manager’s support, the intern context was mostly expressed as an obstacle for municipal employees regarding experimentation opportunities, training, and employer feedback. Indeed, 72.8% said they have little or no training, only 11.6% said they have a lot of opportunities to try new practices in the field of universal accessibility, and 59.7% of the respondents said they have a little or no feedback at all from their employer regarding their abilities implementing universal accessibility measures.

### 3.3. Other Determinants

As 47.6% of participants considered the needs of people with disabilities daily, not at all, or a little, 52.4% considered them moderately or very much. Otherwise, 52.5% of the respondents think that the mandate to meet the needs of those citizens is moderately simple. The impact of social influences did not seem to be consensual. The proportion of participants saying that their colleagues do not believe at all (26.9%) that they should offer services following the municipal action plan is similar to participants saying that their colleagues believe it very much (34.3%). Additionally, similar proportions of participants said they do not have at all (26.0%) or have a very (17.7%) clear idea of how they can provide municipal services that meet the needs of people with disabilities.

### 3.4. CATPCA

Table 2 presents the loadings for each factor related to the 17 variables included in this analysis. The data in bold (loading greater than 0.315) were kept in the analysis. The value of 0.315 squared represents 10% of the explained variance, which correspond to data saturation. In this table, the different determinants are grouped together according to the six factors generated by the factorial analyses. As the implementation of universal accessibility measures is a multidimensional phenomenon, the factors combining determinants influencing each other in the implementation of universal accessibility measures were classified in these six main factors. Six factors were required for the analysis in order to achieve data saturation.

### 3.5. Influencing Factors

Six main factors influencing the determinants emerged from this analysis. These factors were discussed and identified between the authors (MC, FR, SG, JL, and MEL). The first factor overlaps the knowledge and tools that can facilitate the application of universal accessibility measures. Factor 2 corresponds to the context of application, while factor 3 is the integration of the measures in relation to organizational requirements. Factor 4 is the external context that is extrinsic to the person, over which he/she has no control, such as the mandates given to him/her. Factor 5 is the knowledge and skills of the employees towards universal accessibility. Finally, factor 6 groups together the employee’s beliefs about the application of the planned measures. These six factors show us that the context of application (internal) and the knowledge and tools provided are the determinants on which it is most important to act, as they are more clustered together.

## 4. Discussion

Local governments are important actors in implementing actions to meet the CRPD’s expectations, because of their closeness and direct impact on citizens. As such, it is important to look at how different municipalities are taking action to meet the human right of social participation, since the purpose of the CRPD is to promote, protect, and ensure full enjoyment of human rights [1]. Moreover, human rights challenges, such as universal accessibility, and ways to address them need to be planned and are a critical issue at the municipal level [28]. The evaluation of the implementation of universal accessibility principles within the municipal organization’s administration allowed us to better identify the different elements facilitating or hindering the exercise of rights and the response to the needs of people within that environment. Indeed, in a well-designed built environment, a person living with disabilities will experience fewer barriers to social participation and will find it easier to carry out his or her activities independently and to exercise his or her rights fully [29]. The rationale of meeting human rights challenges would ensure planning politics to promote a more inclusive process and better address the universal accessibility measures [28]. The goal of universal accessibility goes beyond reducing discrimination towards people with disabilities; it benefits everyone [5]. Thus, the improvement of knowledge mobilization relevant to universal accessibility in municipalities holds the potential to be beneficial for all citizens and to improve their social participation.

Evidence suggests that universal accessibility increases the quality of life of a majority of individuals and promotes opportunities for social participation and the exercise of their rights on an equal basis with all citizens [3,30,31]. It makes life easier, healthier, and friendlier [5]. The behavior of municipal employees towards the application of universal accessibility has a direct impact on the actualization of these rights for all citizens. Municipal employees do demonstrate a willingness to take action to promote accessible environments and services. While lack of knowledge has a major influence on the ability to act, our results show that it is also important to be able to recognize the rights of others and to understand the impact that accessible environments have on actualizing the rights of people with disabilities.

In this study, we observed that the most facilitating determinant in the application of universal accessibility by the municipal organization’s employees is the beliefs about the impacts of their actions. Regarding the factorial analysis results, the professional and social roles, abilities to implement measures, and access to resources also have an important contribution to the implementation of universal accessibility. When employees believe that their actions have a real impact on citizens with disabilities and the elderly, they are more inclined to adopt good practices. The literature, however, shows that promoting knowledge on the implementation of an innovation is more likely to lead to behavior change [32,33]. This difference may be explained by the fact that studies have evaluated implementation in other types of actions. The studies mentioned above have been conducted in the field of medicine and health. In these fields, knowledge is of primary importance, which may explain why knowledge promotion is the most important determinant. In the case of the implementation of universal accessibility measures, it might be a question of knowing how to behave than of knowledge, which explains why actions are more likely to be implemented when employees understand why they are useful. Implementation authors found that the perception of having learned something appears to be an important facilitator and a precursor to a focused attention and action upon knowledge [32]. However, we find that when individuals are aware of the consequences of their actions in terms of universal accessibility, this increases their ease in implementing the various measures, as mentioned in other studies [21,32,33]. Our study also reveals a disconnection between the knowledge of the theoretical documents and the know-how for the implementation of actions. Although employees do not appear to be familiar with the various documentation and feel that they have few of the required skills, the results do show that municipal employees are able to understand the consequences of their actions on universal accessibility. Changes in behavior and in the organization of services are therefore not conditional on knowledge of the various official documents (e.g., laws, policies, and action plan). It seems even more motivational for municipal employees to implement universal accessibility measures when they know the impacts of their actions on citizens. This non-relevance of knowledge domain is not typical, based on other implementation studies that have used TDF [18,32]. This could be explained by a social desirability bias. Employees may want to show that they want to do well, which makes them feel good about their work. Some may think they have the knowledge, even though this is not really the case. Additionally, employees may not need to know and master the documents and guides in order to be aware of and motivated by the issue of universal accessibility. This could also be due to the organizational or societal culture, which increasingly favors the participation and social integration of all people, including those living with disabilities.

In terms of barriers to the knowledge translation of universal accessibility measures in municipal settings, we have observed that the external context is the most significant barrier. Indeed, the lack of material and financial resources as well as the lack of tools to offer adequate services are important obstacles to the implementation of universal accessibility measures for our participants. Other studies using TDF as a methodological framework showed that external influences, such as access to knowledge and resources, are relevant barriers to implementation [32]. Otherwise, other authors have found that the external context is not a relevant determinant of implementation, while beliefs about their abilities or the consequences of their actions would have an important role to play in the implementation of universal accessibility measures [33]. If we take up Norton’s idea of internal and external determinants, we therefore observed that internal determinants (e.g., beliefs about impacts) are more facilitating than external determinants (e.g., external context).

In some studies using the TDF, social influences have a significant impact on behavioral changes for implementing an innovation [18,34]. This is not the case in this study, as social influences seem to have little impact, either positive or negative, on the implementation of universal accessibility measures by municipal employees. This can be explained by the fact that the impact of social influences depends on the type of organization in which an innovation is implemented. In the case of a municipal organization such as the municipal organization’s administration, the action plans are robust frameworks with defined measures that do not necessarily leave room for external influences, given the various regulations and standards that must be respected.

The TDF was used in numerous research projects related to the implementation of innovations and new practices. Norton, Rodriguez [32] distinguishes internal and external determinants of the TDF. Internal determinants include beliefs about capabilities, consequences, or intentions, whereas external determinants are more environmental contexts or resources. The results of this study show that internal determinants have a greater impact on the implementation of universal accessibility measures by municipal employees, since beliefs about the consequences are what most influence employees’ actions. However, it is important to note the contribution of resources on the ability of employees to implement the various measures in the action plan.

Finally, the factorial analysis allowed to understand what differentiates the change in behavior from one municipal employee to another according to the different determinants of the implementation and to identify on which determinants it is most important to act. Understanding how individuals perceive the context and its setting can provide information on barriers and facilitators to behavioral changes regarding evidence use [33]. CATPCA was used to identify which factors were most important in the employee’s process for the implementation of universal accessibility measures. We observed that the resources, the social and professional role, as well as the capacity to implement such measures according to the level of knowledge and skills were relevant in the implementation of universal accessibility measures, which supports previous research [35].

### Strenghts and Limitations

A first strength of this study is that we received responses from all job types as well as from all administrative units solicited. The participation of the members of the municipal organization advisory committee in the creation of the survey helped to ensure that the questions were adapted to the context. The CAPTCA analysis is also a strength of this study since it brought further analysis to identify the factors influencing the variation of determinants from one employee to the other.

However, while the study has some strengths, it also has limitations. Fatigue might have limited the participants’ capacity to complete the entire survey with the same rigorous method, as some questions required more reflection and thus more time to answer. The 4-level Likert scale used in the survey may have limited the nuances in the participants’ answers. Additionally, the average age of respondents was quite high (41–60 years old), which could produce a generational bias. Indeed, the responses could have been different with younger employees with less experience in the organization but more awareness to this issue. In addition, the use of a survey is less personalized than interviews, for example. This study does not address the need to talk with people about their perceptions but focuses more on the determinants influencing the implementation of universal accessibility measures. Moreover, it is difficult to generalize the results of this study to other municipalities, given the unique and specific context of each municipality. First, the determinants of TDF documented in the survey were identified by the partners in the municipality involved in this study. Thus, it is possible that other determinants would have been documented in other municipalities. Additionally, the environmental context of this Quebec municipality (winter and snow, narrow streets, heritage buildings, and large green spaces) means that the results would probably be different in a municipality where it is always warm or where there is a higher population density, for example. However, we believe that our methodology could be replicated to other municipalities, and that similar organizations could draw useful reflection from our results. 

## 5. Conclusions

This explorative study mostly put forward the influence of the beliefs about the impacts, which indicates that employees are more likely to implement the various measures when they know the concrete impacts of their actions. The importance of the social and professional role and the resources were also confirmed. However, the results show a significant importance of knowledge for employees, as the factor 1 about knowledge and tools that can facilitate the application of universal accessibility measures explains the most variance. This goes hand in hand with studies showing that research and education on universal accessibility is a good first step toward the development of this field [5].

This research allowed us to identify the various factors on which it is possible for this municipality context to take action, particularly with regards to the training of its employees, in order to ensure that the environment allows all citizens to exercise their rights and social participation. The exercise for a municipality, or any other organization, to create an action plan for people with disabilities allows to start a required reflection about the accessibility of the environment. Thus, this research will have allowed the municipal organization to adapt and make a better orientation of the various measures of their new 2021–2027 action plan, in order to facilitate the application of these measures by their municipal employees. The findings of this study and dissemination of these results will provide more information on the particularities of implementation contexts in municipal organizations and the implementation of universal accessibility practices. This research applies a knowledge transfer model in an organizational context and positions the municipality as an actor in the implementation of the best practices of universal accessibility. On one hand, it may help or influence municipal managers’ choices in terms of the best practices of universal accessibility, which could improve the social participation of people with disabilities. On the other hand, it could influence researchers in the field of accessibility to target objectives and solutions to improve implementation practices.

For future directions, while this study has limitations in terms of generalizing the data, we could use these results as a basis for comparing the barriers, facilitators, and determinants of municipal staffing in other urban contexts. It would be interesting to know how barriers or facilitators are different or the same, depending on the geographical position of the municipality. Additionally, there is currently no survey documenting the set of implementation determinants based on the TDF or another implementation model that can meet the needs of organizational contexts. Thus, the creation of an adapted and rigorous survey could be an interesting avenue to standardize the identification of barriers and facilitators to universal accessibility implementation in such a context. Finally, qualitative data collection through interviews, focus groups, or participatory research methods with municipal employees and managers would allow for the implementation of solutions adapted to the needs of municipal employees to overcome barriers to the implementation of universal accessibility.

## Figures and Tables

**Table 1 ijerph-19-14651-t001:** Descriptive analysis of domains of implementation among municipal employees.

Domain	Survey Question Number	Answers	Frequency	Percentage
Knowledge	1	Not at all	20	9.4%
		A little	69	32.5%
		Moderately	81	38.2%
		Very much	42	19.8%
		Total	212	
Skills	2	Not at all	42	21.3%
		A little	59	29.9%
		Moderately	68	34.5%
		Very much	28	14.2%
		Total	197	
Social and professional role	3	Not at all	20	9.7%
		A little	37	18.0%
		Moderately	44	21.4%
		Very much	105	51.0%
		Total	206	
Beliefs in capacities	4	Not at all	21	14.1%
		A little	26	17.4%
		Moderately	78	52.3%
		Very much	24	16.1%
		Total	149	
Beliefs on consequences	5	Not at all	1	0.4%
		A little	4	1.7%
		Moderately	17	7.3%
		Very much	212	90.6%
		Total	234	
	6	Not at all	2	0.9%
		A little	7	3.2%
		Moderately	26	11.9%
		Very much	184	84.0%
		Total	219	
	7	Not at all	17	10.2%
		A little	31	18.6%
		Moderately	56	33.5%
		Very much	63	37.7%
		Total	167	
Intentions	8	Not at all	3	1.9%
		A little	18	11.6%
		Moderately	38	24.5%
		Very much	96	61.9%
		Total	155	
Innovation	9	Not at all	56	26.4%
		A little	45	21.2%
		Moderately	61	28.8%
		Very much	50	23.6%
		Total	212	
	10	Not at all	33	18.0%
		A little	32	17.5%
		Moderately	96	52.5%
		Very much	22	12.0%
		Total	183	
Internal context	11	Not at all	8	5.7%
		A little	20	14.2%
		Moderately	40	28.4%
		Very much	73	51.8%
		Total	141	
	12	Not at all	83	51.2%
		A little	35	21.6%
		Moderately	29	17.9%
		Very much	15	9.3%
		Total	162	
	13	Not at all	56	38.1%
		A little	34	23.1%
		Moderately	40	27.2%
		Very much	17	11.6%
		Total	147	
	14	Not at all	89	59.7%
		A little	27	18.1%
		Moderately	24	16.1%
		Very much	9	6.0%
		Total	149	
External context	15	Not at all	38	23.6%
		A little	49	30.4%
		Moderately	55	34.2%
		Very much	19	11.8%
		Total	161	
	16	Not at all	40	30.5%
		A little	41	31.3%
		Moderately	33	25.2%
		Very much	17	13.0%
		Total	131	
	17	Not at all	35	22.9%
		A little	52	34.0%
		Moderately	38	24.8%
		Very much	28	18.3%
		Total	153	
Social influences	18	Not at all	36	26.9%
		A little	19	14.2%
		Moderately	33	24.6%
		Very much	46	34.3%
		Total	134	
	19	Not at all	47	26.0%
		A little	43	23.8%
		Moderately	59	32.6%
		Very much	32	17.7%
		Total	181	

**Table 2 ijerph-19-14651-t002:** Grouping of the determinants of implementation into six main factors.

	Domains	CATPCA Factors					
		1	2	3	4	5	6
D1	Knowledge	**0.87**	0.20	0.00	0.29	**0.36**	−0.05
D2	Skills	**0.91**	0.14	0.05	0.16	**0.64**	0.10
D3	Social and professional role	0.21	0.20	**0.95**	−0.20	**0.81**	0.20
D4	Beliefs in capacities	0.22	0.14	−0.04	**0.41**	**1.12**	−0.11
D5.3	Beliefs on consequences	0.08	0.00	0.01	0.07	−0.02	**1.21**
D6	Intentions	0.18	−0.05	**1.38**	0.20	−0.13	−0.06
D7.1	Innovation	**1.02**	**0.45**	**0.31**	−0.06	−0.07	0.29
D7.2		**0.66**	−0.06	0.14	**0.78**	−0.08	−0.06
D8.1	Internal context	0.00	**0.48**	0.13	**0.58**	0.14	0.20
D8.2		0.28	**0.86**	−0.02	0.18	0.04	−0.01
D8.3		**0.31**	**0.72**	0.05	0.01	0.02	0.14
D8.4		0.11	**0.90**	0.00	0.17	0.05	−0.05
D9.1	External context	0.14	0.20	−0.03	**0.90**	0.20	0.08
D9.2		0.13	**0.32**	0.00	**0.75**	0.11	−0.06
D9.3		0.00	**0.86**	0.02	**0.32**	0.17	−0.09
D10.1	Social influences	**0.44**	**0.60**	0.00	0.12	0.07	0.04
D10.2		**0.81**	**0.44**	0.10	0.23	−0.01	−0.08

Numbers in bold are >0.30 and so support the factors explaining the variance and influencing the determinants.

## Data Availability

Not applicable.

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
