# Peer review of "Implementation Determinants of Knowledge Mobilization within a Quebec Municipality to Improve Universal Accessibility"

_ijerph, 2022, doi:10.3390/ijerph192214651_

Round 1

Reviewer 1 Report

The article deals with an interesting topic. Paper has the potential, however, in order to be published, it needs to be improved.

The authors have not justified the need for conducting proposed research (authors have not elaborated on what are the underlying theoretical issues that need to be answered), and no gap was identified. The overall literature review provides a very brief, weak, non-coherent, and unconvincing story. The literature review section requires major rewriting for a research article. 

Although your work analyses some interesting propositions, the scope of your article, in terms of the data used, appears limited. Consequently, I am concerned that your submission does not seem to provide a generalizable contribution to public health practice. The paper needs to provide a significant contribution beyond the specific setting analyzed in your work.

However, the authors should consider publishing a revised paper if they (1) more clearly describe the scope and significance of their paper (authors should provide an answer to WHY the findings are important and worthy of dissemination), (2) justify the selection process in terms of the data and sample used, and (3) provide a stronger contribution ("theoretical and managerial implications"), future research directions and discuss the generalizability and limitations of the study.

Reviewer 2 Report

I am grateful for the possibility to become familiar with this manuscript.

This article is well researched, argumented and written. I would only recommend to maybe revise the abstract to make it smoother /more readable. The present abstract does seem to follow the IMRAD scheme, however, when searching in a database, the abstract is a basis for our decision to download the paper, read it and – maybe - quote it. So, the authors should consider making it look more organic, while still following the IMRAD scheme (to briefly, concisely and clearly summarize basic goals, methods, results and conclusions of the study undertaken in the article).

Good luck!

Round 2

Reviewer 1 Report

The paper can be accepted.